# New Perspectives in the Management of Chronic Hand Eczema: Lessons from Pathogenesis

**DOI:** 10.3390/ijms25010362

**Published:** 2023-12-27

**Authors:** Vittorio Tancredi, Dario Buononato, Stefano Caccavale, Eugenia Veronica Di Brizzi, Roberta Di Caprio, Giuseppe Argenziano, Anna Balato

**Affiliations:** Dermatology Unit, Department of Mental and Physical Health and Preventive Medicine, University of Campania Luigi Vanvitelli, 80131 Naples, Italydario.buononato@gmail.com (D.B.);

**Keywords:** chronic hand eczema, dermatitis, irritant contact dermatitis, allergic contact dermatitis, atopic dermatitis

## Abstract

Chronic hand eczema (CHE) is a common inflammatory skin condition that significantly impacts the quality of life. From work-related disabilities to social embarrassment, pain, and financial costs, the burden on society is substantial. Managing this condition presents challenges such as long-term treatment, poor patient compliance, therapy side effects, and economic feasibility. As a result, significant efforts have been made in this field in recent years. Specifically, the broader understanding of CHE pathogenesis has led to the development of new drugs, both topical and systemic. The aim of this narrative review is to summarize the current available data on hand eczema pathophysiology and explore the resulting developments in drugs for its treatment. A comprehensive search on PubMed and the other main scientific databases was conducted using keywords related to CHE and its pathogenesis. The most relevant pathways targeted by therapies include the JAK-STAT cascade, IL-4, and IL-13 axis, phosphodiesterase 4 enzyme, and chemo-attractant cytokines. In the near future, physicians will have a plethora of therapeutic alternatives. Consequently, they should be well-trained not only in how to use these alternatives but also how to combine these treatments to address the ongoing challenges related to efficacy, tolerability, and safety.

## 1. Introduction

Hands are one of the most socially and functionally significant organs in the human body. Touching, greeting, and working are primary social functions typically carried out by hands. Therefore, their effectiveness and appearance hold great importance for people. It is not surprising that hand diseases can compromise many aspects of daily life, significantly impacting overall quality of life.

Chronic hand eczema (CHE) is a relatively common skin condition, and while it is not life-threatening, it can be quite distressing when it reaches a moderate-to-severe form [1]. It is also one of the most common occupational diseases, often leading to sick leave. Some reviews have estimated the economic burden of CHE, which includes costs for prescriptions, medications, laboratory exams, and productivity losses [2]. This burden is likely underestimated, as it appears to affect approximately 5–10% of the population, with a similar prevalence among adolescents and adults [1]. Skin examination may reveal a wide range of clinical manifestations characterized by erythema, vesicles, pustules, scales, fissures, and hyperkeratosis (Figure 1).

While the diagnosis is generally not difficult, there can be some pitfalls. Differential diagnoses include tinea manum, scabies, mycosis fungoides, hand-foot-mouth disease, and, above all, palmar psoriasis (PP) [1]. Diagnosing psoriasis can be challenging, particularly when dealing with the hyperkeratotic variant of chronic hand eczema (CHE). In such cases, a thorough personal and family history and a comprehensive examination to detect other psoriatic lesions are useful. Dermoscopy can be helpful, too. Detection of white scales is typical for PP, while the presence of yellowish scales, brownish-orange dots/globules and yellowish-orange crust is more suggestive for CHE [3]. In difficult cases, an incisional biopsy with histological examination is required. Risk factors for CHE include frequent hand washing, exposure to wet conditions, concurrent atopic dermatitis, and occupational and not occupational exposures to various substances [1,4]. These substances can be found in cosmetics, jewelry, clothing, and medications and are often work-related [5]. Three main subcategories are recognized: irritant contact hand dermatitis, allergic contact dermatitis, and atopic hand dermatitis [1]. These three entities should not be considered separate entities but are deeply interconnected. However, some key differences in immune responses have been studied. Irritant contact dermatitis (ICD) is characterized by a predominance of innate immune system responses. Allergic contact dermatitis (ACD) is classically associated with a delayed type IV acquired immune reaction involving both CD4+ and CD8+ T lymphocytes, with specific Th1 differentiation [6]. Atopic eczema has a complex pathology, with a predominant Th2-type response associated with skin water loss, cutaneous barrier defects, and dryness [7].

Conventional treatments include both topical and systemic medications. Topical treatments are based on steroids, calcineurin inhibitors, vitamin D derivate, and emollients. Systemic drugs include steroids, retinoids, and immunosuppressants such as cyclosporine and azathioprine. Currently, the only systemic drug approved for moderate-to-severe CHE is alitretinoin, which belongs to the retinoid class [8]; in fact, retinoids administered systemically improve CHE clinical course, while topical treatment with retinoids leads to a worsening of the TEWL.

CHE often poses a challenge for both patients and physicians. A definitive treatment does not exist, and the chronic course of the condition can lead to poor compliance with topical products and side effects with systemic drugs. The increasing knowledge about the pathophysiology and immunology of CHE is paving the way for the development of new and promising drugs. Some of these treatments are already available, while others are still in phase II/III clinical trials. The aim of this review is to summarize our understanding of hand dermatitis pathogenesis and how this knowledge contributes to the management of this condition.

## 2. Material and Methods

Our work is a narrative review. We performed research on the principal scientific databases of PubMed Central^®^ (PMC), and the U.S. National Institutes of Health’s National Library of Medicine (NIH/NLM). The research was conducted with the keywords “contact irritant dermatitis”, “allergic contact dermatitis”, “atopic dermatitis”, “chronic hand eczema”, “chronic hand dermatitis”, “hand dermatitis”, and “hand eczema”. Furthermore, we carried out a second research study with the combinations of previously mentioned keywords and the words relevant to the pathogenesis and treatment of chronic hand eczema, like “JAK-STAT”, “IL-4”, and “IL-13”. Particularly, we focused on the most recent drugs available for CHE. Only English works were evaluated. Only articles with titles and abstracts pertinent to our review were included. Of a total of 148 results, only 59 articles were included in this review. No restrictions of pertinent data were preliminarily performed, but in cases of repetition of concepts or data, only more recent results were selected.

## 3. Results

### 3.1. Classic and New Models of Contact Dermatitis

The pathophysiology of irritant and allergic dermatitis has historically been divided into non-immunologic and immunologic models. Initially, irritant contact dermatitis (ICD) was believed to be directly caused by the toxic action of an irritant substance (depending on its own chemical-physical nature) without immune system involvement. However, subsequent research has revealed the critical role of the innate immune response in ICD. The penetration of chemicals induces the production of inflammatory cytokines by keratinocytes, such as Tumor Necrosis Factor α (TNFα) and arachidonic acid derivatives, leading to the recruitment of various immune cells like neutrophils, mast cells, and macrophages. This results in skin damage, necrosis, and inflammation [9]. On the other hand, allergic contact dermatitis (ACD) is characterized by a delayed type IV adaptive immune response. Previous exposures to an antigen can lead to the selection of specific T lymphocytes (sensitization phase), which are promptly activated in subsequent encounters (elicitation phase) and produce various cytokines like interferon γ (IFN-γ)*,* IL-2, and IL-17. These cytokines are responsible for keratinocyte apoptosis, inflammation, and eczema [10]. For many years, ACD was thought to be primarily a Th1 reaction, but more recent studies have shown that different allergens can stimulate different types of T-cell responses [11]. For example, fragrances mainly induce the expression of Th2 genes [6]. However, this dichotomy is overly simplistic, as it is now understood that the two immune systems are deeply interconnected. In ACD, the sensitization phase involves antigen presentation by antigen-presenting cells (APCs) to naïve lymphocytes. This antigen presentation depends on the innate immune system, as it is essential for an efficient specific immune response [12]. Atopic dermatitis, a significant risk factor for CHE, has a complex pathophysiology with the predominant involvement of Th2 cells and certain cytokines, such as IL-4 and IL-13, associated with skin barrier defects [13].

### 3.2. Conventional Treatments

Conventional treatment of CHE can be divided into topical and systemic drugs (Table 1). Topical comprises emollient, steroid, and calcineurin inhibitors. The first line recommendation always includes the avoidance of allergic substances or irritants. Nearby, all substances can act as irritants in the appropriate concentration and if contact lasts for the right amount of time. Allergens instead require patch tests to be individuated. However, even if the suspected agents are successfully avoided, inflammation can become chronic. Emollients should always be prescribed for CHE management. They provide lipids like ceramides, whose concentration is often altered in these conditions. Thus, the emollient participates in the repair of disrupted skin barriers, which would facilitate the penetration of these exogenous substances [14]. Topical steroids are a milestone in the management of CHE. They are available in different formulations and potency. The most powerful ones, like clobetasol and triamcinolone, can allow remission from the acute phase in a few weeks, but their application cannot be chronic. Continuous steroid administration leads to steroid-dependent dermatitis, skin atrophy with consequent worsening of eczema, and systemic absorption [15,16]. Calcineurin inhibitors like pimecrolimus and tacrolimus are other topicals that are approved for the management of AD and can be effective in CHE. While not associated with important side effects like topical steroids, they seem to be insufficient to adequately control hand eczema [17]. Calciprotriol is a vitamin D3 derivative that is usually prescribed for vulgar psoriasis. It has also been evaluated in CHE, and some studies reported results similar to steroids. The main side effects are represented by dryness and irritation [17]. Phototherapy can also be evaluated. In particular, local psoralen plus UVA (PUVA) seems to be more efficient than narrow-band UVB [18]. The marketing of new drugs has made phototherapy more and more a second option for its logistic complications. Of the systemic therapies, the most prescribed are steroids and retinoids. Alitretinoin is currently the only systemic drug approved for CHE in different countries, including Europe. It is a vitamin A derivative that targets the retinoic acid receptors A and X. It acts on keratinocyte proliferation, angiogenesis, and apoptosis and has immune-modulatory properties. 30 mg of alitretinoin was shown to be effective in the amelioration of CHE versus placebo; however, side effects like teratogenicity, dryness, cheilitis, myalgia, headache, suicidality, and serum lipids alteration do not make it a convincing drug for long course treatment [19]. In a 2018 meta-analysis evaluating up to 3734 patients, alitretinoin led to clean hands in about 50% of participants, particularly with higher doses for a long period [20]. Steroids, instead, are rarely administered, usually for flare-up management. They are contraindicated due to several chronic side events [17]. There are other immunosuppressants that have been studied, in particular cyclosporine, mycophenolate mofetil, azathioprine, and methotrexate. The first is an inhibitor of T-cell response by targeting the calcineurin protein, while the others target the DNA synthesis. Data available are represented only by case reports, and no blind randomized trials have been carried out with these drugs [21].

### 3.3. New Treatments

Progress in science has opened our knowledge of both new pathogenic patterns and new technologies. These elements are revolutionizing the approach to different diseases, from psoriasis to atopic dermatitis, contact dermatitis, and CHE, too. Here, we examined the most recent findings in CHE physiopathology and how these change our therapeutic arsenal (Table 1).

#### 3.3.1. JAK-STAT Pathway

Janus Kinase (JAK) proteins are membrane cellular receptors that play a key role in the signal transduction of various mediators. They are divided into four groups with their own specific cascade and terminal gene expressions: JAK1, JAK2, JAK3, and TYK2. STAT proteins are associated with JAK molecules, and after receptor activation, they can migrate to the cell nucleus and induce the transcription of different genes. In total, seven different variants of STAT proteins have been identified. The JAK-STAT pathway is responsible for the effects of several cytokines involved in many diseases. Due to the wide spectrum of cytokines dependent on these receptors, the JAK pathway is crucial in the differentiation of Th1, Th2, Th17, and Th22 lymphocytes [22]. As mentioned earlier, contact dermatitis and atopic hand dermatitis are usually correlated with Th1 and Th2 lymphocytes, respectively. Thus, there is a rationale for employing this knowledge in CHE treatment. For these reasons, in the last decades, many efforts have been carried out to develop drugs specifically targeting the JAK-STAT pathway. Systemic JAK inhibitors can be less manageable than other drugs like dupilumab. The induction phase usually requires a comprehensive laboratory check, with a particular focus on blood count, lipid profile, cardiovascular risk factors, and major infectious diseases like viral hepatitis and tuberculosis. Apart from hypersensitivity, other absolute contraindications are represented by pregnancy, severe hepatic disease, end-stage kidney disease, and tuberculosis [23]. Physicians should also keep in mind that JAK inhibitors can interact with other drugs, particularly those metabolized by CYP3A4 [24]. Eventually, the European Medicines Agency (EMA) and the Food and Drug Administration (FDA) have emphasized that systemic JAK inhibitors (like upadacitinib, tofacitinib, etc.) should not be the first choice for patients over 65 years or those at risk for cardiovascular, thromboembolic, and neoplastic events, although the overall risk is low [23,24].

##### Delgocitinib

Delgocitinib is a pan JAK inhibitor that modulates the Th2 response, acting on several cytokines such as IL4, 6, 22, and others. It is one of the most studied drugs for the management of CHE. Topical ointment was compared to a placebo in a phase 2 trial concluded in 2016. Significant improvement was detected in the group that applied delgocitinib ointment compared to the placebo [25]. Positive results were gained in all CHE subtypes. Side events were reported only in a minority of patients, and they mainly consisted of mild upper airway infections (17.3–29.4% in delgocitinib groups vs. 40% in other groups) and headaches (3.8–11.5% in delgocitinib groups vs. 4.0% in other groups). A more recent 2b phase trial investigated the efficacy of delgocitinib cream compared to a placebo in 258 patients. Delgocitinib cream 8 and 20 mg/g was revealed to be superior to the vehicle (*p* < 0.001) in Investigator’s Global Assessment for CHE (IGA-CHE) achievement at week 16. They also showed a statistically significant improvement in hand eczema severity index (HECSI), itch and pain numerical rating scale (NRS) scores, and Patient’s Global Assessment (PGA) compared to the placebo at week 16 [26]. The side effects collected were concordant with the previous trials. In particular, nasopharyngitis was observed in 7.3–29.4% of delgocitinib groups vs. 40% of the vehicle group, eczema occurred in 5.8–11.3% of the delgocitinib groups as opposed to 16.0% in the vehicle group, and head pain was experienced by 3.8–11.5% in the delgocitinib groups and 4.0% in the vehicle group [26].

##### Ruxolitinib

It is an inhibitor of JAK 1 and 2 receptors. It has been employed in the management of different hematopoietic dyscrasia (like myelofibrosis and polycythemia) for many years [27,28]. As a topical treatment, it has been evaluated in several dermatologic diseases such as atopic dermatitis, alopecia areata, vitiligo, and psoriasis [29]. In atopic dermatitis, results are promising. In recent phase 3 studies, significantly more patients achieved IGA treatment response with 0.75% ruxolitinib cream (50.0%/39.0%) and 1.5% ruxolitinib cream (53.8%/51.3%) versus vehicle (15.1%/7.6%; *p* < 0.0001) at week 8 [30]. Thus the drug seemed to be promising also for CHE. An investigator-initiated, open-label, single-site study (NCT05293717) is currently evaluating the efficacy of 12 weeks of ruxolitinib 1.5% treatment in CHE. The results available after 4 weeks showed that the drug is efficient in reducing IGA (79% had ≥ 2 points reduction) and HECSI (100% reached HECSI 50 while 64% reached HECSI 75). An important reduction of pruritus was detected as well. No side effects were reported [31]. Nevertheless, randomized double-blind trials are necessary to further investigate the efficacy, tolerability, and side effects of this drug in CHE.

##### Gusacitinib

Gusacitinib is an oral small molecule that blocks the pathway of spleen tyrosine kinase (SYK) and Janus kinases (JAK 1, 2, 3, and TYK2), managing to inhibit the signaling of Th17, Th1, Th2, and Th22 cells [32]. As with other systemic JAK inhibitors, it shows the same class-related side events such as anemia, lymphopenia, an increase of triglycerides, cholesterol, and risk of thromboembolism. A recent phase 2b study evaluated the efficacy of gusacitinb in 97 patients with CHE recalcitrant to other conventional therapies. Patients who randomly received 40 or 80 mg of gusacitinb experienced significant improvement versus the placebo. Precisely, at week 16, patients assuming 80 mg gusacitinib obtained a 69.5% (*p* < 0.005) reduction in the modified total lesion-symptom score versus 49.0% for 40 mg (*p* = 0.132), and 33.5% for placebo. Important enhancement Physician’s Global Assessment (PGA) was observed in 31.3% of patients receiving 80 mg versus 6.3% of the placebo (*p* < 0.05). A 73.3% decrement in the hand eczema severity index versus placebo (21.7%) was seen in patients receiving 80 mg (*p* < 0.001). No significant side effects were described; these were mainly represented by upper airway infection, headache, and nausea. Five patients had serum creatine phosphokinase elevation [33].

##### Upadacitinib

Upadacitinib is a selective oral JAK 1 inhibitor currently approved for the management of moderate-to-severe atopic dermatitis (AD). Preliminary trials, Measure Up 1 and Measure Up 2, demonstrated significant improvements in HECSI scores from baseline with both 15 and 30 mg doses of upadacitinib [34]. These promising results were subsequently confirmed in real-life databases. In the BioDay registry, 32 patients with atopic dermatitis and comorbid contact hand eczema were followed. A HECSI-75 score was achieved by 59.3% of patients, and 74.1% achieved (almost) clear skin, as assessed by the photographic guide [35]. Acne was the most common adverse reaction encountered in the preliminary trials. In most cases, it was mild and easily manageable with topical drugs. Other commonly reported sides effects were upper airway infections (11.4/100 upadacitinib 30 mg vs. 12.2/100 upadacitinib 15 mg), urinary tract infection (5.3/100 upadacitinib 30 mg vs. 12.2/100 upadacitinib 15 mg), herpes zoster (4.6/100 upadacitinib 30 mg vs. 3.4/100 upadacitinib 15 mg), oral herpes (7.1/100 upadacitinib 30 mg vs. 3.3/100 upadacitinib 15 mg), and increased creatine phosphokinase (11.1/100 upadacitinib 30 mg vs. 7.0/100 upadacitinib 15 mg). A non-fatal stroke and two thromboembolic events occurred during the trial, but they were judged not related to upadactinib [36].

##### Baricitinib

Baricitinib is a selective, reversible oral inhibitor of JAK1 and JAK2, with a lower affinity for JAK3 and TYK2. It is currently approved for the treatment of alopecia areata and atopic dermatitis [37]. Dosages of 2 mg and 4 mg are available. Reported adverse events are similar to other JAK inhibitors like upadacitinib [38]. Two case reports have outlined positive responses in 11 patients with contact hand and foot eczema [39,40].

##### Abrocitinib

Abrocitinib is an oral JAK1 inhibitor currently available for moderate-to-severe atopic dermatitis. In phase 3 studies, JADE MODO I and JADE MODO II, abrocitinib 100 and 200 mg results were superior to the placebo. In particular, at week 12, about 40% and 60% of patients treated with abrocitinib 100 and 200 mg, respectively, reached Eczema Area and Severity Index (EASI) 75. Instead, EASI 75 was obtained by only 10% of patients who assumed a placebo [41]. Data extrapolation from the phase 3 study JADE DARE showed that a major proportion of patients with atopic dermatitis and concomitant CHE treated with abrocitinib 200 mg achieved healed hands compared to those treated with dupilumab 300 mg [42].

#### 3.3.2. Phosphodiesterase 4 and Cyclic Adenosine Monophosphate in Skin Inflammation

Cyclic adenosine monophosphate (cAMP) is an intracellular compound that plays a key role in regulating cell metabolism, signal transduction, and gene expressions [43]. Specifically, it induces the expression of the anti-inflammatory cytokine IL-10. This mediator reduces the levels of pro-inflammatory cytokines such as TNFα and INFγ, which appear to be involved in CHE and several other diseases like psoriasis. Consequently, drugs targeting cAMP have been the subject of recent studies. Notably, cAMP is degraded by phosphodiesterase 4 (PDE4). Effectively, inhibiting PDE4 increases cellular cAMP concentration. As a result, inhibitors of PDE4 have been synthesized. Currently, there are systemic PDE4 inhibitors like apremilast and topical inhibitors like roflumilast and crisaborole [44].

##### Apremilast

Apremilast is a systemic inhibitor of PDE4. It is labeled for moderate-to-severe forms of psoriasis, psoriasic arthritis, and behçet syndrome. Apremilast is indicated in patients who have severe comorbidities like cancer and important infectious diseases [45]. Different studies remark on its efficacy as an off-label treatment in other dermatologic disorders like atopic dermatitis [46]. In a case report, the authors reported a complete resolution of a CHE after one month of apremilast 30 mg/twice a day in patients affected by hepatic cirrhosis [47].

##### Roflumilast

Roflumilast is an inhibitor of phosphodiesterase 4 (PDE4). In the dermatological field, it has been mainly employed as a topical for plaque psoriasis treatment [48]. For atopic dermatitis, roflumilast cream 0.5% has been evaluated versus placebo in a randomized, vehicle-controlled phase 2a trial. No statistically significant amelioration in EASI score was observed, while significant relief of itching occurred according to the NRS pruritus scale [49]. Regarding CHE, a multicenter, double-blinded, randomized, placebo-controlled, clinical trial is currently evaluating the efficacy and safeness of roflumilast 500 mcg tablet versus placebo [50]. Preliminary data are still not available.

##### Crisaborole

Crisaborole is a topical inhibitor of PDE4. It has proved its efficacy in atopic dermatitis both in adults and children. Additionally, a retrospective review involving 251 patients indicates that it could be a successful therapy for CHE, with 72.2% of individuals experiencing an amelioration in their symptoms [51].

#### 3.3.3. IL-4/13 Axis

IL-4 and IL-13 have focused researchers’ attention in the last years. These two cytokines have a fundamental role in the Th2 response, whose primary role is to face parasitic and helminthic infections. IL-4 has been found to be crucial in the differentiation and sustenance of this lymphocyte family. Moreover, IL-4 induces the IgE switch of B lymphocytes. Instead, IL-13 mainly participates in the activation and recruitment of eosinophil cells. They both participate in the process of tissue healing [52]. More recently, an important role of this axis has been discovered in atopic and allergic phenomena. In fact, proteins critical for maintaining the integrity of the skin barrier, including involucrin, loricrin, and filaggrin, experience downregulation due to the overexpression of IL-4 and IL-13. This downregulation results in epidermal thinning, increased permeability to environmental antigens, and subsequent inflammation of the skin [13].

##### Dupilumab

Dupilumab is a fully human antibody directed against the common alpha chain of IL-4 and 13 receptors, two key players in atopic dermatitis with their role in modulating skin barrier integrity. It is currently marked for moderate-to-severe atopic dermatitis. In adults, the usual posology consists of 600 mg at time 0 by subcutaneous injections followed by 300 mg every 2 weeks [53]. Some studies proved its efficacy also in the management of CHE. The effectiveness of dupilumab in atopic hand eczema may be explained because some allergens like fragrances and balsam of Perù are specifically active in the Th2 pathway [6]. Some reports show that dupilumab was effective in 162 atopic hand dermatitis and 72 contact hand dermatitis patients with different subtypes of CHE who failed other conventional treatments. Hyperkeratotic CHE seems to be the most resistant subtype to dupilumab [54]. Dupilumab usually has a favorable security and tolerability profile, with no really serious side effects or drug interactions. The most known adverse reactions are the ocular ones, like dupulimab-associated conjunctivitis (8.2%) [55]. On the other hand, it is quite expensive and should not be prescribed as first line.

##### Tralokinumab

Tralokinumab is a human antibody targeting IL-13 specifically, without involvement of IL-4. It showed superiority compared to placebo in bettered EASI and IGA; thus, it is currently approved for moderate-to-severe atopic dermatitis [56]. The security profile is very favorable and similar to dupilumab, but today, there are few data about its efficacy in CHE. A case report described a patient with atopic dermatitis and concomitant hand eczema who positively responded to tralokinumab in the first seven months of therapy. Unfortunately, the patient developed psoriasis, probably due to a switch from a Th2 response to a Th1 one, and the drug was interrupted [57].

#### 3.3.4. Chemo-Attractant Cytokines in Contact Dermatitis

Recent studies have highlighted the role of the chemokine CCL2 and its primary receptor, CCR2, which is highly expressed by skin macrophages. External irritants can induce the production of CCL2 by keratinocytes, and its binding to CCR2 leads to the expression of IL-1β, a cytokine responsible for neutrophil recruitment and which has a central implication in inducing inflammation [58].

##### AFX5931

It is a small molecule targeting the CCL5 and CCL2. These cytokines seem mainly involved in the initial phases of ICD and ACD. Topical formulation has been evaluated versus placebo in a phase 4 study with 15 patients, but results are still not published [59].

## 4. Conclusions

CHE continues to pose significant diagnostic and management challenges for both patients and physicians. Moreover, it constitutes a noteworthy socioeconomic issue. Therefore, establishing an optimal therapeutic strategy is of paramount importance. Conventional treatments, even those officially endorsed, often fall short of ensuring favorable outcomes. In particular, alitretinoin is the only approved and the most studied in this field, but albeit effective, there are issues about chronic employment side effects and the prolonged teratogenic risk in fertile women. However, recent advancements in both technology and our understanding of the underlying pathogenic mechanisms have opened new avenues for CHE management. Beyond the specific subtypes of CHE, several common therapeutic targets have emerged, including the JAK-STAT pathway, IL-4/13 axis, PDE4, and chemo-attractive chemokines. Among these, JAK inhibitors stand out as one of the most promising drug classes. They are available in both topical and systemic forms, with clinical trials demonstrating their efficacy. The most evaluated in CHE are delgocitinib as a topical and gusacitinib as a systemic drug. Main challenges arise when considering the long-term safety of these drugs, particularly in light of declarations from regulatory bodies like EMA and FDA. Dupilumab, a prominent therapy targeting IL-4 and IL-13, has demonstrated well-documented efficacy, tolerance, and safety in atopic dermatitis. However, our knowledge in the context of CHE, especially in forms unrelated to atopy, remains limited. While we acknowledge that certain allergens can notably stimulate the Th2 response, questions persist regarding the suitability of dupilumab for other subtypes of CHE. Other targets have not yet provided well-organized data. Open problems concern irritant and allergic subtypes. In this case, avoidance of the substance involved is the best preventive and curative treatment. In conclusion, CHE is more and more a matter of concern. The increasing number of drugs that will probably be available will open new issues about the possible combination of treatments and their efficacy, tolerability, and safeness (a treatment algorithm is provided in Figure 2).

## Figures and Tables

**Figure 1 ijms-25-00362-f001:**
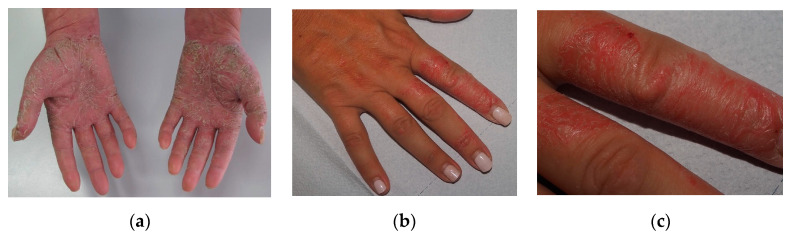
Clinical presentation of chronic hand eczema. (**a**) Chronic hand eczema with important lichenification of the palms. (**b**) Hand eczema in the exudative phase with erythema, blisters, and erosions on the back of the fingers. (**c**) In detail hand eczema of the second and third fingers of the right hand.

**Figure 2 ijms-25-00362-f002:**
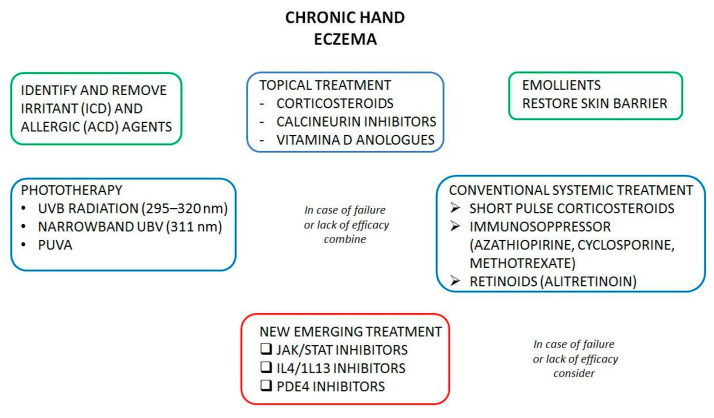
Treatment algorithm in Chronic Hand Eczema. Green, behavioral and precautionary measures. Blue, current therapeutic options. Red, emerging treatment.

**Table 1 ijms-25-00362-t001:** Main systemic and topical treatments for chronic hand eczema classified by target or class.

Target Pathway or Class	Systemic	Topical
Corticosteroids	Systemic corticosteroids (prednisone, others)	Topical corticosteroids (clobetasol, others)
Vitamin D		Calcipotriol
Retinoids	Alitretinoin	
Calcineurin inhibitors		Pimecrolimus Tacrolimus
JAK/STAT	Abrocitinib Baricitinib Gusacitinib	Delgocitinib Ruxolitinib
	Upadacitinib	
IL-4/IL-13	Dupilumab (IL-4/IL-13)	
	Tralokinumab (IL-13)	
PDE4	Apremilast	Crisaborole
		Roflumilast
CCL5/CCL2	AFX5931	

## Data Availability

Not applicable.

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
