# Peer review of "New Perspectives in the Management of Chronic Hand Eczema: Lessons from Pathogenesis"

_ijms, 2023, doi:10.3390/ijms25010362_

Round 1
Reviewer 1 Report
Comments and Suggestions for Authors
The manuscript by Balato et al. submitted for publication in IJMS is a review describing the most recent findings in chronic hand eczema (CHE) physiopathology and exploring the developments in drugs for its treatment. The authors considered the most relevant pathways targeted by therapies which include the JAK-STAT cascade, IL-4 and IL-13 axis, phosphodiesterase 4 enzyme and 21 chemo-attractant cytokines.
Overall the review is well structured but I suggest that authors should consider making further enhancements before this manuscript is considered for publication in IJMS. These comments are listed below:
-It would be more appealing to insert some figures that summarize the concepts
-The acronym, for e.g. CHE, must only be reported the first time in the extensive form
Comments on the Quality of English LanguageEnglish language only needs to be slightly implemented.
Author Response
The authors would like to thank Editor for careful revision of our manuscript and providing us with its comments and suggestion to improve the quality of the manuscript. The following responses have been prepared to address of the editor’s comments.

Reviewer 2 Report
Comments and Suggestions for Authors
Line 81: It should be differentiated here that topical treatment with retinoids leads to a worsening of the TEWL. Only systemically administered retinoids do not show this
Line 92: The platforms should be precisely identified
Line 208: How frequent was the occurrence of these symptoms?
Line 235: Stating that JAK1, JAK2, JAK3 and TYK2 are addressed.
Line 239: The phrase "and so on" should be avoided. Either refer to a previous complete list of side effects or list them here.
Lines 260-263: The frequencies would be interesting here
Line 262: did the authors mean "stroke" here?
Line 268: Spelling error in JAK
Line 278: I would recommend a different description of healed hands instead of clean hands.
Lines 292/293: The point of the sentence doesn't make sense to me
Line 303: How was the sensation of pruritus measured?
Line 333: What was the exact clinical background of the 162 and 72 patients?
Line 337: Frequency?
Line 358: Study number?
Line 374: misspelling of "systemic"
Minor linguistic improvements necessary (spelling).
Author Response

(The authors gave the same response as above.)

Reviewer 3 Report
Comments and Suggestions for Authors
-
The narrative review effectively delineates the current landscape of chronic hand eczema treatment, specifically highlighting the utilization of target-specific molecules. It adeptly synthesizes the most recent knowledge available, showcasing an in-depth understanding of the subject matter.
-
However, in its pursuit of presenting established information, the review inadvertently overlooks the integration of novel perspectives or emerging viewpoints. By incorporating diverse viewpoints, the review could achieve a more comprehensive outlook, potentially enriching its content.
-
Given its fundamental objective of elucidating chronic hand eczema treatment through the lens of pathophysiology, the review would greatly benefit from augmenting its textual narrative with visual aids. Illustrations, tables, or diagrams could serve as valuable supplements, facilitating a deeper grasp of the intricacies involved in treatment methodologies.
-
The inclusion of these visual elements would not only bolster the review's comprehensibility but also enhance its appeal to readers within the Journal's audience. Such enhancements could significantly elevate the accessibility and engagement level of the content, making it more impactful and informative.
Author Response
The authors would like to thank Reviewer for careful revision of our manuscript and providing us with its comments and suggestion to improve the quality of the manuscript. The following responses have been prepared to address of the reviewer's comments.

Round 2
Reviewer 3 Report
Comments and Suggestions for Authors
The insertion of the new Figure 2 has significantly enhanced the revised version of the manuscript, providing a more visually appealing representation of its contents. Additionally, a minor correction is needed: the wavy red underlines for the word "lack" should be removed from Figure 2.